# Multi-Omics Characterization of Early- and Adult-Onset Major Depressive Disorder

**DOI:** 10.3390/jpm12030412

**Published:** 2022-03-06

**Authors:** Caroline W. Grant, Erin F. Barreto, Rakesh Kumar, Rima Kaddurah-Daouk, Michelle Skime, Taryn Mayes, Thomas Carmody, Joanna Biernacka, Liewei Wang, Richard Weinshilboum, Madhukar H. Trivedi, William V. Bobo, Paul E. Croarkin, Arjun P. Athreya

**Affiliations:** 1Department of Molecular Pharmacology and Experimental Therapeutics, Mayo Clinic, Rochester, MN 55901, USA; grant.caroline@mayo.edu (C.W.G.); wang.liewei@mayo.edu (L.W.); weinshilboum.richard@mayo.edu (R.W.); 2Department of Pharmacy, Mayo Clinic, Rochester, MN 55901, USA; barreto.erin@mayo.edu; 3Department of Psychiatry and Psychology, Mayo Clinic, Rochester, MN 55901, USA; kumar.rakesh@mayo.edu (R.K.); skime.michelle@mayo.edu (M.S.); 4Department of Psychiatry and Behavioral Sciences, Duke University, Durham, NC 27701, USA; rima.kaddurahdaouk@duke.edu; 5Department of Medicine, Duke University, Durham, NC 27708, USA; 6Duke Institute for Brain Sciences, Duke University, Durham, NC 27710, USA; 7Department of Psychiatry, Peter O’Donnell Jr. Brain Institute, University of Texas Southwestern Medical Center, Dallas, TX 75235, USA; taryn.mayes@utsouthwestern.edu (T.M.); madhukar.trivedi@utsouthwestern.edu (M.H.T.); 8Department Population and Data Sciences, University of Texas Southwestern Medical Center in Dallas, Dallas, TX 75390, USA; thomas.carmody@utsouthwestern.edu; 9Department of Quantitative Health Sciences, Mayo Clinic, Rochester, MN 55901, USA; biernacka.joanna@mayo.edu; 10Department of Psychiatry and Psychology, Mayo Clinic, Jacksonville, FL 32224, USA; bobo.william@mayo.edu

**Keywords:** genomics, metabolomics, major depressive disorder, age at onset, network analysis

## Abstract

Age at depressive onset (AAO) corresponds to unique symptomatology and clinical outcomes. Integration of genome-wide association study (GWAS) results with additional “omic” measures to evaluate AAO has not been reported and may reveal novel markers of susceptibility and/or resistance to major depressive disorder (MDD). To address this gap, we integrated genomics with metabolomics using data-driven network analysis to characterize and differentiate MDD based on AAO. This study first performed two GWAS for AAO as a continuous trait in (a) 486 adults from the Pharmacogenomic Research Network-Antidepressant Medication Pharmacogenomic Study (PGRN-AMPS), and (b) 295 adults from the Combining Medications to Enhance Depression Outcomes (CO-MED) study. Variants from top signals were integrated with 153 p180-assayed metabolites to establish multi-omics network characterizations of early (<age 18) and adult-onset depression. The most significant variant (*p* = 8.77 × 10^−8^) localized to an intron of *SAMD3*. In silico functional annotation of top signals (*p* < 1 × 10^−5^) demonstrated gene expression enrichment in the brain and during embryonic development. Network analysis identified differential associations between four variants (in/near *INTU, FAT1, CNTN6,* and *TM9SF2)* and plasma metabolites (phosphatidylcholines, carnitines, biogenic amines, and amino acids) in early- compared with adult-onset MDD. Multi-omics integration identified differential biosignatures of early- and adult-onset MDD. These biosignatures call for future studies to follow participants from childhood through adulthood and collect repeated -omics and neuroimaging measures to validate and deeply characterize the biomarkers of susceptibility and/or resistance to MDD development.

## 1. Introduction

Major depressive disorder (MDD) etiology and prognosis vary by the age at depressive onset. Early onset is characterized by poorer quality of life, greater psychiatric and medical comorbidity, higher heritability, and increased suicidality [1,2,3,4,5]. This suggests that individuals with early-onset MDD may benefit from a tailored pharmacologic treatment approach [6,7,8,9,10]. However, psychotherapy and pharmacotherapy remain largely consistent across the age spectrum [11]. Before pharmacotherapy can be individualized according to the age at MDD onset, deeper characterizations of the biological differences between early and later-onset presentations are necessary.

Several genome-wide association studies (GWAS) and one exome-wide association study have been performed to understand the genomics of MDD specific to age at onset [8,12,13,14,15]. Collectively, and in conjunction with polygenic risk analyses, these studies suggest that earlier-onset MDD may share greater genetic overlap with schizophrenia, bipolar disorders, and attention deficit/hyperactivity disorder (ADHD) than later onset [8,12,16,17]. Two GWAS investigating the age at depressive onset (*N* = 2746 [14]) and (*N* = 9238 [13]) did not identify overlapping top signals (*p* < 1 × 10^−5^), highlighting phenotypic heterogeneity that may require additional biological measures to understand. Circulating metabolite concentrations are highly heritable (up to 62%) [18,19] and are differentially perturbed in MDD according to age [10], yet the genetic factors associated with these differences have not yet been investigated. To our knowledge, no integrative genomic and metabolomic approaches have been undertaken to identify multi-omics signatures which best characterize and differentiate patients with early- and late-onset depression. Such an analysis may yield insights into heritable (genomic) and downstream (metabolomic) differences between age groups.

Multi-omics strategies are amongst the approaches at the forefront of individualized medicine in psychiatry and are grounded in the understanding that complex traits (e.g., onset of depression) cannot be fully characterized by isolate biomeasures [20,21,22,23,24,25,26,27,28]. Rather, interactions between genes, transcripts, proteins, metabolites, and the environment determine the development and trajectory of complex diseases [29]. Multi-omics strategies can aide in biomarker selection for future experimental validation by identifying the features which best characterize conditions of interest (e.g., early- vs. adult-onset MDD) [30]. This is imperative in this context given the breadth of the metabolomic and genomic platforms. The objectives of this study were therefore to (1) identify risk and protective single nucleotide variants (SNVs) associated with age at onset of MDD using a GWAS approach, and (2) integrate genomic findings with plasma metabolites to identify multi-omics differences between individuals with early- (<age 18) versus adult-onset MDD. We hypothesized that this multi-omics approach would derive distinct molecular signatures of early- versus adult-onset MDD.

## 2. Materials and Methods

### 2.1. Data Sources

This was a cross-sectional, secondary analysis of Caucasian adults with MDD from the PGRN-AMPS (*N* = 486 with genomics; *N* = 245 with multiple omics) and CO-MED (*N* = 295 with genomics; *N* = 76 with multiple omics) studies (Table 1; see Appendix A for sample inclusion). The PGRN-AMPS (NCT00613470) and CO-MED (NCT00590863) trials enrolled outpatients with moderate to severe nonpsychotic MDD. These trials have been characterized in detail in prior publications [31,32]. The CO-MED study included patients who met DSM-IV-TR criteria for recurrent or chronic MDD (current episode lasting ≥ 2 years), while PGRN-AMPS included patients with DSM-IV diagnosed MDD, without the requirement of recurrent or chronic presentation. The CO-MED cohort also represents a more regionally diverse population—the trial was conducted across six primary and nine psychiatric care sites (compared to the single-site PGRN-AMPS trial). The inclusion and exclusion criteria were otherwise similar between studies. Both studies were conducted in accordance with the approval of their respective Institutional Review Boards. Informed consent was obtained from all participants.

### 2.2. Measures and Outcomes

Age at depressive onset was self-reported with a baseline case report form in PGRN-AMPS and with the Mini-International Neuropsychiatric Interview (MINI) [7] in CO-MED. Baseline depression severity was assessed with the 16-item Quick Inventory of Depressive Symptomatology—Clinician-Rated (QIDS-C) [33]. Race and ethnicity were self-reported and validated through genomic techniques (described in Genotyping, Genome-Wide Imputation, and Quality Control).

### 2.3. Genotyping, Genome-Wide Imputation, and Quality Control

PGRN-AMPS: Genotyping was performed at the RIKEN Center for Genomic Medicine (Yokohama, Japan) using Illumina human 610-Quad BeadChips (Illumina, San Diego, CA, USA), with quality control procedures and imputation as previously described [26,31,34]. Following genotyping, quality control, and imputation, 7,017,931 variants meeting a minor allele frequency threshold of 0.01 and an imputation R^2^ > 0.3 in 486 Caucasian individuals were retained for statistical analysis.

CO-MED: Genotyping was performed using Illumina Quad, Human Omni 2.5 bead chips, as previously described [35]. During quality control, samples were removed if self-reported sex was incongruent with X-chromosome estimated sex, call rate was <98%, or heterozygosity ratio was outside the mean (≤0.7 on any chromosome). One individual from each pair determined to have kinship > 0.08 by the King-Robust test was removed. Concordance between intended duplicates was assessed. Variants were removed if unmapped, duplicated, achieved call rate < 95% or achieved Hardy Weinberg equilibrium *p* < 1 × 10^−10^. These criteria resulted in 2,356,856 variants from 464 individuals available for phasing and imputation. Imputation was performed using the Michigan Imputation Server [36] with EAGLE2 and the Haplotype Reference Consortium (HRC) reference panel (version r1.1 2016) for phasing [37]. Data were aligned to the human reference genome GRCh37. Following quality control and imputation, 8,286,413 variants meeting a minor allele frequency threshold of 0.01 and an imputation R^2^ > 0.3 in 295 Caucasian individuals were retained for statistical analysis. Race was self-reported and concordant with genetic clustering based upon multidimensional scaling [38].

### 2.4. Genome-Wide Association Study Statistical Analyses

GWAS for age at depressive onset as a continuous trait was performed with univariate linear regression under an additive genetic model [39]. GWAS was performed separately in the PGRN-AMPS and CO-MED cohorts to assess the replicability of signals across independent cohorts. Prior to analysis, age at depressive onset was log2 transformed to meet the assumption of normality. Analyses were adjusted for sex, and the first ten principal components of ancestry and filtered for minor allele frequency >5%. Quantile-quantile plots of expected vs. observed *p*-values were constructed to assess for statistical inflation (Appendix A). Given that the purpose of GWAS was to identify candidates for subsequent multi-omics integration analyses, index variants associated with age at depressive onset *p* < 1 × 10^−5^ were retained. Index variants were defined as the variant with the lowest *p*-value amongst all associated variants (r^2^ > 0.8) in each signal. Sensitivity analyses were conducted excluding individuals with self-reported depressive onset prior to age three, as evidence suggests that clinically significant depression can arise as early as age three [40,41,42,43]. Analysis was performed in PLINK version 2.0 (Mountain View, CA, USA) [44], and Manhattan plots were created with LocusZoom.org [45].

### 2.5. Variant Annotation

All variants meeting suggestive significance were annotated with the nearest gene, location to nearest gene, and consequences (e.g., non-synonymous, missense, synonymous variant). Next, these variants and variants in linkage disequilibrium (r^2^ > 0.8) were assessed for known cis- and trans- expression quantitative trait loci (eQTL) gene labels. Gene annotation was done with the HaploReg database [46], implemented in R studio with haploR [47]. Following variant annotation, the nearest genes and eQTL genes for the 36 index SNVs were input into the Database for Annotation, Visualization and Integrated Discovery (DAVID) [48] v6.8 (Frederick, MD, USA) for tissue-specific expression analysis, tissue-associated protein enrichment, and pathway and disease over-representation.

### 2.6. Metabolomics

Plasma metabolites were assayed in a subset of PGRN-AMPS and CO-MED patients using the AbsoluteIDQ p180 platform [49]. This targeted assay utilizes triple quadrupole tandem mass spectrometry to detect metabolites from five analyte classes (acylcarnitines, amino acids, biogenic amines, glycerophospholipids and sphingolipids). This assay is not exhaustive of all potential metabolite correlates of early onset depression, and it does not measure several metabolites which are well-studied in psychiatry (i.e., dopamine, GABA). However, these analytes have enhanced our understanding of the biology of MDD [20,50,51,52], schizophrenia [53], and psychosis [54,55], warranting their investigation in the context of early-onset MDD. Metabolomic profiling and quality control were conducted separately for the PGRN-AMPS and CO-MED cohorts, as described in previous publications [51,52,56]. Metabolites with ≥10% missingness were excluded from the current analysis, leaving 153 metabolites available for analysis (Appendix A). All metabolites meeting quality control criteria were included in the current integration analysis to identify which may be candidates for future, targeted studies.

### 2.7. Multi-Omics Integration Network Analysis

The purpose of the current multi-omics integration network analysis is to (a) compare differences in individuals with early (<age 18) vs. adult onset MDD and (b) identify biomarkers capable of characterizing such phenotypes.

Inputs to the analysis included the 36 index SNVs from GWAS signals with suggestive association with age at depressive onset (*p* < 1 × 10^−5^ in either cohort) and 153 p180-assayed metabolites meeting quality control criteria. The threshold of suggestive significance was defined in accordance with several GWAS studies and the GWAS catalog [57]. These metabolites represent five classes (glycerophospholipids, amino acids, biogenic amines, acylcarnitines, and sphingolipids), and are synthesized and metabolized by enzymes encoded in the host genome. These metabolic classes are amongst those which are essential for growth, development, and many key physiological functions [58].

The analysis utilizes a sparse partial least squares discriminant approach [59] to identify significant correlations between SNVs and metabolites (*p* < 0.05; |r| > 0.1) in individuals with early and adult onset MDD. Furthermore, the analysis includes a multi-level community detection step [60] to identify communities (sub-networks) of SNVs and metabolites which are strongly associated within their community and less associated outside of the community. Community detection accounts for the correlation structure amongst metabolites when assessing metabolite-variant associations. The assumption underlying community detection is that communities are comprised of functionally related biomolecules [30].

The analysis was performed using xMWAS (Atlanta, GA, USA) [30] as implemented in R v4.0.3 (Vienna, Austria) [61] with RStudio version 1.3 (Vienna, Austria) [62]. Results were visualized using Cytoscape (Bethesda, MD, USA) [63].

## 3. Results

### 3.1. Sample Characteristics

Table 1 provides the demographic characteristics for the samples. Generally, the sample characteristics for the two studies were similar, although the CO-MED sample had a higher ratio of Hispanic participants than PGRN-AMPS. Mean age at baseline was younger in those with early onset than adult onset (37.3 ± 12.7 vs. 43.8 ± 12.8; *p* < 0.05); all other baseline characteristics were similar between early- and adult-onset groups.

### 3.2. Genome-Wide Association Study Statistical Analyses

GWAS for age at depressive onset was performed in the PGRN-AMPS and in CO-MED cohorts (Manhattan plots illustrated in Figure 1). No SNVs met genome-wide statistical significance after accounting for multiple comparisons (*p* < 5 × 10^−8^) in either cohort. However, a signal of three SNVs in tight linkage-disequilibrium (LD) in an intron of the Sterile Alpha Motif Domain Containing 3 (*SAMD3)* gene (chromosome 6) achieved near statistical significance in the CO-MED cohort, with the tag SNV (rs870816) reaching *p* = 8.8 × 10^−8^ (*t* = −4.49). The negative direction of the t-statistic effect size for rs870816 signifies that the minor allele associates with earlier MDD onset. There was no overlap between cohorts in the signals which achieved suggestive significance (*p* < 1 × 10^−5^). Fifty-five SNVs met suggestive significance (*p* < 1 × 10^−5^) in PGRN, and 79 SNVs met suggestive significance in CO-MED (Appendix A). One individual in the CO-MED study reported depressive onset prior to three years of age (self-reported age = 0) and was excluded in a subsequent GWAS sensitivity analysis. The top signals found in the sensitivity analysis remain consistent with the analysis which includes this individual (Appendix A). Of these total 134 SNVs with suggestive significance, minor alleles for 91 of them associated with earlier MDD onset (negative effect size), while 54 associated with later onset (positive effect size). The 36 index SNVs for these signals are shown in Table 2. Genomic control λ values were within 1 ± 0.05 for each analysis.

### 3.3. Variant Annotation

Twenty of the 36 index SNVs localized to introns; the remaining SNVs localized outside of the open reading frames. Given that no SNVs localized within coding regions, there were no non-synonymous or missense variant candidates. Eleven variants are known eQTLs for one or more genes in one or more tissues (Table 2). Functional annotation of the genes using DAVID [48] at a nominal *p*-value (*p* < 0.05) demonstrated that the tissue-associated protein enrichment for these genes was greatest in the brain (“UP_TISSUE” annotation). Furthermore, the tissue-specific expression analysis (“UNIGENE EST QUARTILE” annotation) most strongly highlighted tissues involved in embryonic development. Multiple psychiatric traits are also represented at *p* < 0.05, including tobacco use disorder and alcohol use disorder (Appendix A).

### 3.4. Multi-Omics Integration Network Analysis

Integration analysis identified two communities of correlated (*p* < 0.05; |r| > 0.1) SNVs and metabolites in individuals with early (<age 18) onset MDD (Figure 2A), and three communities in individuals with adult onset MDD (Figure 2B). Four SNVs in total were represented (locus zoom plots, see Appendix A). Notably, the SNV nearest *INTU* correlated with four phosphatidylcholine diacyl species (34:1, 36:4, 36:5, 36:6) in individuals with early-onset MDD, and 11 phosphatidylcholine species, (nine of which are acyl-alkyl species), in individuals with adult-onset (Figure 2). Early onset was also characterized by a strong positive correlation between glutamine and rs2793779 near *TM9SF2*, which is absent from the adult-onset network. Individuals with adult onset MDD demonstrate unique correlations between the (*FAT1)* intronic SNV and phosphatidylcholines, spermidine, histidine, tryptophan, and three lysophosphatidylcholine metabolites. Finally, several carnitines correlated with the SNV closest to *CNTN6* in individuals with adult onset MDD but not in individuals with early-onset MDD (Figure 2). Pearson correlations for all SNVs and metabolites represented in these networks can be found in Appendix A.

## 4. Discussion

This work establishes multi-omics characterization of early- versus adult-onset MDD using data from the PGRN-AMPS and CO-MED studies. GWAS for age at depressive onset were performed for both PGRN-AMPS and CO-MED, and top variants were then integrated with plasma metabolomics to identify biological signatures which best differentiated early and adult-onset MDD. These multi-omics networks enabled biological characterization of MDD by age at onset and provide a basis for future functional experiments aiming to investigate the mechanisms underlying the development of MDD across the lifespan. Derived early- and adult-onset MDD biosignatures showed distinct associations between variants in/near *INTU, FAT1, CNTN6,* and *TM9SF2* with plasma metabolites (phosphatidylcholines, carnitines, biogenic amines, and amino acids) for continued future investigation.

Expression of top variants identified through GWAS for age at depressive onset was enriched in the brain and during embryonic development according to the DAVID database, which provides functional annotations for lists of variants [48]. No variant achieved genome-wide significance, although this is not uncommon for GWAS for psychiatric traits given the phenotypic complexity [8,13,64,65,66]. Valuable mechanistic insights into biological drivers of neuropsychiatric diseases and drug response have been gained from GWAS signals of suggestive significance from the PGRN-AMPS cohort, despite its limited sample size [26,67,68]. Several variants meeting suggestive significance (mapping to *SAMD3*, *GRIN2B)* have previously been implicated in MDD [69,70], with our current study adding to this body of literature by revealing novel associations with age at MDD onset. In several instances, the minor allele of GWAS variants associated with a later onset of MDD, suggesting they might be protective against disease onset. Alternatively, these variants may confer MDD susceptibility by interacting with risk factors specific to later life, for example, functional/cognitive impairment or marital challenges [71]. In this context, continued investigation of the biological implications of top GWAS signals through multi-omics integration analysis was warranted.

The multi-omics integration analysis characterized the variant-metabolite associations that best differentiated early (<18 years) and adult-onset MDD. Onset of MDD was dichotomized at age 18 for this assessment for several reasons. For many, the transition at age 18 into legal adulthood includes major changes in determinants of health such as financial resources and social environments [72,73,74]. Clinically, age 18 also corresponds to a transition from adolescent to adult medical care. Furthermore, most research studies use age 18 to define early versus adult samples. Finally, age 18 was the median age at MDD onset in CO-MED and near the median (20 years) in PGRN-AMPS, so this threshold enabled a relatively balanced sample split.

The multi-omics networks demonstrated that rs1399212, nearest the “Inturned Planar Cell Polarity Protein (*INTU*)” gene, negatively associated with phosphatidylcholines of both networks, and had unique associations with the phosphatidylcholine acyl-alkyl (PC-ae) species in individuals with adult-onset MDD. The ether phosphatidylcholines (PC-ae species) play distinct roles from conventional phosphatidylcholines (PC-aa species) in cell differentiation, cell signaling, and reduction of oxidative stress [75]. These network associations suggest that *INTU* variation may impact concentrations of phosphatidylcholine metabolites, which is corroborated by associations of *INTU* and trans fatty acid levels in a sample of >9000 individuals with replication across diverse ancestries [76]. Furthermore, the differential *INTU*-phosphatidylcholine associations in early vs. adult-onset MDD, with adult-onset having many PC-ae associations, suggest that regulation of phosphatidylcholine biosynthesis or metabolism by *INTU* may be biologically distinct between groups, potentially conferring effects on cell differentiation, cell signaling, and reduction of oxidative stress.

INTU protein is a central component of the CPLANE (ciliogenesis and planar polarity effector) module [77], which is critical for ciliogenesis and transport of ciliary proteins [77,78,79] (Figure 3). Cilia are highly conserved organelles which form specialized extensions of the cell membrane, and they may be motile or non-motile. In the brain, motile cilia are present on the ependymal cells of the ventricles and in the choroid plexus, whereas non-motile cilia are widely observed, including on astrocytes, neurons, and progenitors [80]. Cilia are critical for embryonic development [81,82,83,84]. Both during development and in adult homeostasis, cilia facilitate cellular signal transduction along major signaling pathways, including Hedgehog, Wnt, Notch, transforming growth factor B, G protein-coupled receptors, receptor tyrosine kinases, extracellular matrix receptors [85,86,87,88], pathways that are known to be perturbed in MDD [89,90]. Perturbations in cilia structure and/or function can lead to a spectrum of diseases (‘ciliopathies’) with broad developmental and adult phenotypes, including cognitive defects and neuropsychiatric phenotypes [91,92]. The cilia interactome, defined as interactions between ciliary proteins, also demonstrates extensive overlap with neuropsychiatric disease interactomes and genes differentially expressed in neuropsychiatric diseases, including MDD, schizophrenia, bipolar disorder, Alzheimer’s disease, ADHD, autism spectrum disorder, and Parkinson’s disease [93,94]. At the macroscopic level, *INTU* variants are amongst the polygenic signals associating with cortical thickness (5 × 10^−9^) in children of the Adolescent Brain Cognitive Development (ABCD) Study [95]. Cortical thickness, in turn, is dysregulated in MDD compared with controls [96,97,98], and a preliminary study suggests differences in cortical thickness between pediatric and adult onset MDD [99].

The differential associations of specific phosphatidylcholine species and *INTU* variants according to age at depressive onset may be indicative of a mechanistic basis distinguishing early and adult-onset MDD. These associations may be mediated by additional biological and environmental factors (Figure 3), and they may have a functional relationship with the observed cortical thickness differences in patients with MDD across the lifespan. Given the role of *INTU* in neurodevelopment, whether these differences are specific to the neurodevelopmental period or whether they persist into adulthood remains a question for future investigation. Based on these networks, further characterization of the precise lipid content within ciliary membranes and their trafficking may help advance our biological understanding of early versus adult-onset MDD.

The multi-omics networks also reveal additional associations for future mechanistic studies. This includes the associations of variation near *CNTN6* with plasma carnitines, which are specific to adult-onset MDD. Such an investigation will add to the current characterizations of subgroups of MDD by acylcarnitine metabolomic profiles [50]. This also includes the associations of variants in *FAT1* with spermidine, given that spermidine has recently been proposed as an antidepressant drug [100,101]. Perhaps the most critical future direction arising from this work’s identification of differential variant-metabolite associations in early and adult-onset MDD will be for future studies to assay biomeasures across the lifespan to enhance our understanding of the development of MDD. The present data were sourced from adults. A characterization across the lifespan of biochemical (e.g., omics) and neuroimaging measures is needed, given the suggested interplay between *INTU*, plasma metabolomics, and cortical thickness in differentiating individuals by the age at MDD onset. Except in the case of rare mutation, an individual’s genome remains consistent throughout the lifespan. Meanwhile, the regulation of genomics via downstream biological and environmental factors creates a dynamic metabolome. Epigenetics may be particularly valuable in future studies for explaining the differential gene-metabolite associations in early versus adult-onset depression. Early life adversity can confer epigenetic modifications linked to development of depression in youth and adults [102]. Epigenomic alterations may drive neuropsychiatric disease through changes in gene expression and neural function, and individuals with adult versus later onset depression have been successfully discriminated by genome-wide DNA methylation markers [103,104]. Integrating biomeasures (e.g., -omics including epigenomics) in children and older adults alongside environmental (e.g., stress) information will further clarify mechanisms of MDD development at various stages of life and may enable strategies to mitigate risk.

While these findings are novel, this study acknowledges limitations. The top GWAS signals were not genome-wide significant and did not overlap in the PGRN-AMPS and CO-MED studies despite comparable age at depressive onset, sex, and baseline depression levels upon study enrollment. Differences may be attributed to ethnicity differences across cohorts (higher proportion of Hispanic individuals in CO-MED), enrollment criteria differences (PGRN-AMPS patients included all MDD, while CO-MED patients specifically had chronic or recurrent MDD), and/or limited sample sizes to investigate these hypotheses. Findings should therefore be validated in larger cohorts. Age at depressive onset was self-reported without independent verification. The multi-omics signatures of early and adult-onset MDD were derived from samples of adults with MDD. The extent to which these signatures replicate in the context of child and adolescent patient samples must be evaluated. Ideally, as MDD is not a static disorder but rather a disorder that may recur and change, longitudinal studies should exist which collect measures (e.g., metabolomics, neuroimaging) throughout various stages of health and disease to assess the trajectory of the illness. This may also enable identification of ‘hidden nodes’ underlying genomic-metabolomic associations to derive insights into their functional relationships. PGRN-AMPS and CO-MED did not uniformly collect data on environmental contributors to MDD or age-specific risk factors including puberty or menopause. Therefore, the extent of environmental stressors or hormonal influences were unaccounted for in the analyses. This study was performed in Caucasian individuals to maximize sample size and to avoid confounding of genomic findings from differences in minor allele frequency across ancestries. The characterized network signatures should be replicated in larger cohorts and across ancestries to assess the generalizability of these findings. These studies lacked uniform data on the duration of depressive illness and the number of prior depressive episodes at the time of metabolomic assay. Metabolites were not collected in a systematic manner (e.g., under fasting conditions, at a uniform time of day), which may confer noise to the analyses, although fasting status may not significantly impact laboratory variability for most metabolites [105]. The p180 platform does not assay several key psychiatric neurotransmitters, including dopamine and GABA, and the platform was unable to detect serotonin at the quality control threshold employed. Lipids assayed by current mass spectrometry technology may actually reflect sum signals of all isomeric/isobaric compounds having the same parent and daughter ions [49]. Therefore, future studies should validate the identified lipid metabolites with additional assays.

In conclusion, this study identified candidate variants through genome-wide analyses which associate with the development of MDD across the lifespan. Novel multi-omics integration analysis with top GWAS variants and plasma metabolomics enabled characterization of biosignatures of early and adult onset MDD. Such networks serve two purposes: First, they enable hypothesis generation for future mechanistic studies of the development of MDD throughout the lifespan. For example, this may include follow-up investigation of *INTU*-phosphatidylcholine functional relationships. Second, they demonstrate potential differential genomic regulation of the plasma metabolome by the age at MDD onset. The results here encourage future longitudinal studies to collect and integrate additional -omics (e.g., proteomics, exposomics) and neuroimaging at multiple timepoints to enable deeper biological characterization of MDD. This may ultimately help parse the heterogeneity of MDD and enable insights into biological drivers and protectors of the disease across the lifespan.

## Figures and Tables

**Figure 1 jpm-12-00412-f001:**
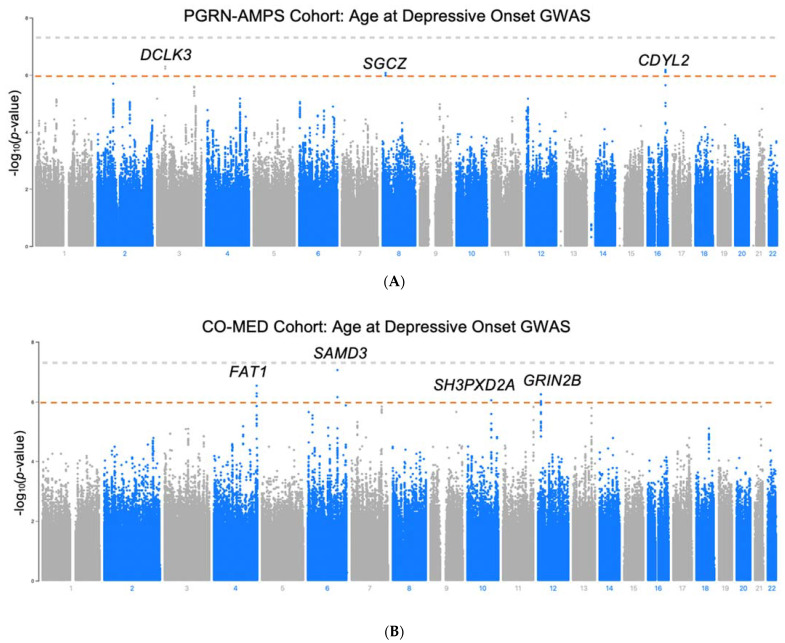
Manhattan plots for GWAS for age at depressive onset in (**A**) PGRN-AMPS cohort (*N* = 487) and (**B**) CO-MED (*N* = 295). Grey line: genome-wide significance (5 × 10^−8^). Red line: suggestive significance (1 × 10^−6^).

**Figure 2 jpm-12-00412-f002:**
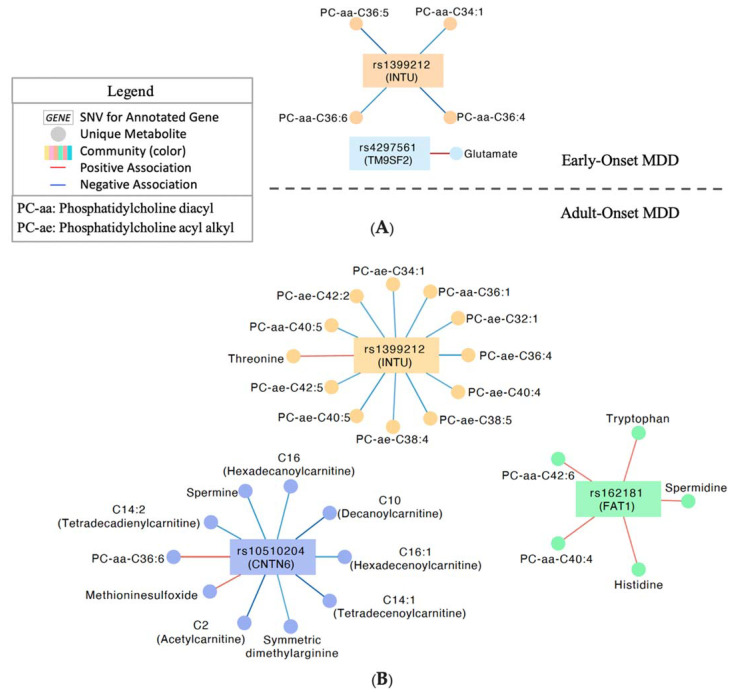
Multi-Omics integration network analysis. (**A**) Significant (*p* < 0.05; |r| > 0.1) associations of SNVs and metabolites in individuals with depressive onset prior to age 18. (**B**) Significant (*p* < 0.05; |r| > 0.1) associations in individuals with depressive onset at or after age 18. Legend applies to both (**A**) and (**B**).

**Figure 3 jpm-12-00412-f003:**
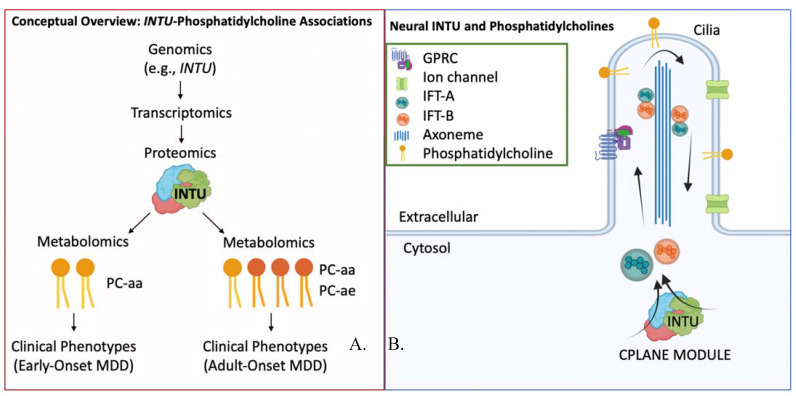
Analysis conclusions and future aims. (**A**) Modulators (e.g., transcriptomics and proteomics) may mediate the differential associations between variation near INTU and phosphatidylcholine metabolites in early vs. adult-onset MDD. Future investigation into such modulators may enhance our understanding of the development of MDD at various stages across the lifespan. (**B**) INTU is a member of the CPLANE protein module which facilitates intraflagellar transport of proteins and lipids throughout the cilia via the intraflagellar transport A and B (IFT-A, IFT-B) complexes. Phosphatidylcholines are lipids widely present in the lipid membrane which facilitate receptor localization in the membrane and signaling via intracellular cascades.

**Table 1 jpm-12-00412-t001:** Demographics and patient characteristics.

	Genomics Analyses		Multi-Omics Analyses	
	CO-MED	PGRN-AMPS	Early Onset (<18 Years of Age)	Adult Onset (≥18 Years of Age)
Total (N)	295	486	130	191
PGRN-AMPS (N)	-	486	100	145
CO-MED (N)	295	-	30	46
Age [mean (SD)]	43.2 (12.9) *	39.9 (13.7) *	37.3 (12.7) *	43.8 (12.8) *
Ethnicity (% Hispanic)	20% *	2.1% †*	4.8% †	9.2% †
Sex (% Women)	68%	62%	71%	62%
Age at depressive onset (Median [min, max])	18 [0, 60]	20 [4, 83]	14 [4, 17] *	33 [18, 83] *
Baseline depressive severity [mean (SD)]	15.7 (3.4)	15.1 (3.5)	15.3 (3.6)	14.8 (3.0)

* Significantly different (*p* < 0.05) by Fisher’s Exact test or independent-samples *t*-test (calculated for all variables excluding study Ns). † Ethnicity characterizations calculated from a smaller subset of samples based upon data availability (117 individuals with early depressive onset; 177 individuals with adult depressive onset; 338 PGRN-AMPS individuals).

**Table 2 jpm-12-00412-t002:** Index variants by study.

A. PGRN-AMPS GWAS Index Variants											
Variant ID	SNV	Minor Allele	Nearest Gene	Distance to Nearest Gene	Location	eQTL	Minor Allele Frequency	Beta	Standard Error	T-Statistic	*p*-Value
1-90283515-A-G	rs7545243	G	*LRRC8D*	3056		RP5-943J3.1 (subcutaneous adipose, whole blood, lung), LRRC8B, LRRC8C, LRRC8D (whole blood)	0.29	−0.25	0.06	−4.54	7.16 × 10^−6^
2-71442773-G-C	rs59432780	C	*PAIP2B*	0	Intron	PAIP2B (skin)	0.12	−0.38	0.08	−4.81	2.01 × 10^−6^
2-71456834-T-C	rs56796378	C	*PAIP2B*	0	Intron	MPHOSPH10 (whole blood)	0.10	−0.40	0.09	−4.53	7.48 × 10^−6^
2-144491308-G-A	rs10928197	A	*ARHGAP15*	17,222			0.43	−0.23	0.05	−4.49	8.8 × 10^−6^
2-144477195-G-A	rs12465492	A	*ARHGAP15*	3681			0.38	−0.24	0.05	−4.48	9.43 × 10^−6^
3-36755107-G-A	rs35721771	A	*DCLK3*	610,866			0.35	0.28	0.06	5.09	5.16 × 10^−7^
3-1006912-A-G	rs10510204	A	*CNTN6*	613,027			0.40	0.24	0.05	4.55	6.75 × 10^−6^
4-150429619-G-A	rs6853045	G	*DCLK2*	569,805	Intron		0.46	−0.24	0.05	−4.56	6.67 × 10^−6^
6-5389403-C-A	rs73350538	A	*FARS2*	0	Intron		0.06	0.51	0.11	4.49	8.79 × 10^−6^
8-14827133-A-C	rs76522180	C	*SGCZ*	0	Intron		0.06	−0.57	0.11	−4.99	8.43 × 10^−7^
12-9058993-A-T	rs7299653	T	*PHC1*	8321		KLRG1 (subcutaneous and visceral omentum adipose, aorta and tibial artery, brain (cortex), mammary tissue, fibroblasts, esophagus, atrial appendage, tibial nerve, pancreas), PHC1 (esophagus), RP11-436I9.6 (lung), RP11-705C15.3 (skeletal muscle), LINC00987 (subcutaneous adipose, whole blood), RP11-118B22.4 (whole blood), M6PR (whole blood), A2MP1 (whole blood)	0.48	−0.24	0.05	−4.55	6.76 × 10^−6^
16-80882476-T-C	rs9926993	C	*CDYL2*	44,300			0.07	−0.51	0.10	−5.04	6.72 × 10^−7^
16-80842302-A-T	rs62052150	T	*CDYL2*	4126			0.08	−0.46	0.10	−4.48	9.57 × 10^−6^
**B. CO-MED GWAS Index Variants**											
3-93956553-T-A	rs143801763	A	*NSUN3*	110,922			0.08	0.63	0.14	4.54	8.27 × 10^−6^
3-104317729-G-A	rs4450851	A	*MIR548A3*	371,623			0.44	−0.34	0.07	−4.55	8.10 × 10^−6^
4-187604429-T-C	rs162181	C	*FAT1*	0	Intron		0.20	−0.48	0.09	−5.25	2.96 × 10^−7^
4-187592223-T-A	rs11723473	A	*FAT1*	0	Intron		0.16	−0.45	0.09	−4.78	2.83 × 10^−6^
4-128523964-T-C	rs1399212	C	*INTU*	30,121			0.06	−0.71	0.15	−4.59	6.64 × 10^−6^
6-130472920-C-T	rs870816	T	*SAMD3*	0	Intron	L3MBTL3, SAMD3 (whole blood)	0.34	−0.40	0.07	−5.49	8.77 × 10^−8^
6-130474597-T-G	rs1932106	G	*SAMD3*	0	Intron	SAMD3 (whole blood)	0.45	0.38	0.07	5.08	6.94 × 10^−7^
6-166957334-T-C	rs6456092	C	*RPS6KA2*	0	Intron		0.37	0.37	0.07	4.95	1.32 × 10^−6^
6-5367273-G-A	rs797147	A	*FARS2*	0	Intron		0.11	−0.55	0.11	−4.83	2.21 × 10^−6^
6-21867547-A-G	rs6940645	G	*FLJ22536*	0	Intron		0.38	−0.36	0.08	−4.78	2.86 × 10^−6^
6-89987882-G-T	rs12213221	T	*GABRR2*	0	Intron		0.47	−0.34	0.07	−4.57	7.45 × 10^−6^
6-130499476-C-T	rs1034263	T	*SAMD3*	0	Intron	SAMD3 (whole blood)	0.07	−0.67	0.15	−4.55	7.82 × 10^−6^
7-132382054-C-T	rs10233511	T	*PLXNA4*	48,606	Intron		0.29	−0.40	0.08	−4.92	1.45 × 10^−6^
7-26421269-T-G	rs74409431	G	*SNX10*	7319		AC004540.4, SNX10 (subcutaneous adipose)	0.13	0.51	0.11	4.66	4.80 × 10^−6^
9-114014832-T-C	rs112014566	C	*OR2K2*	74,929			0.11	−0.59	0.12	−4.84	2.19 × 10^−6^
10-105498155-T-A	rs7085238	A	*SH3PXD2A*	0	Intron	SH3PXD2A (whole blood)	0.22	−0.44	0.09	−5.03	8.86 × 10^−7^
11-134403616-C-A	rs12287910	A	*LOC283177*	28,060		GLB1L2|B3GAT1 (whole blood)	0.42	0.36	0.07	4.92	1.48 × 10^−6^
12-13883111-C-T	rs11609779	T	*GRIN2B*	0	Intron		0.18	0.50	0.10	5.12	5.61 × 10^−7^
12-13880168-G-C	rs10845837	C	*GRIN2B*	0	Intron		0.30	0.39	0.08	4.83	2.24 × 10^−6^
13-100223130-G-A	rs4297561	A	*TM9SF2*	7853			0.33	0.39	0.08	4.90	1.61 × 10^−6^
13-100224460-G-A	rs2793779	A	*TM9SF2*	9183			0.33	0.38	0.08	4.77	2.97 × 10^−6^
18-55933912-C-T	rs62094545	T	*NEDD4L*	0	Intron		0.17	0.44	0.10	4.55	7.86 × 10^−6^
21-40135684-T-A	rs8127960	A	*NCRNA00114*	0	Intron	LINC00114 (tibial artery)	0.25	0.43	0.09	4.93	1.44 × 10^−6^

## Data Availability

All raw and analyzed data and related materials, including programming code, are available upon request to Mayo Clinic. PGRN-AMPS data have been deposited on dbGaP, study accession phs000670.v1.p.1.

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
