# Peer review of "Multi-Omics Characterization of Early- and Adult-Onset Major Depressive Disorder"

_jpm, 2022, doi:10.3390/jpm12030412_

Round 1
Reviewer 1 Report
This paper pinpoints an emerging need to consider genomics together with metabolomics when studying complex diseases such as MDD. However, the "multi-omics" analysis described in the manuscript is far from an accurate multi-omics scale analysis. The so-called "multi-omics" analysis in the manuscript is merely a correlation detection process between 36 SNVs and 153 metabolites using a network approach (plus community detection) performed by a bioinformatics tool. This is at most another level of functional association analysis based on the GWAS results. I admire the authors' ambitions to do omics data integration, and I do believe this is a good strategy in the post-GWAS era. Still, I would go for semantic accuracy first and prefer a less ambitious title for this paper.
Other suggestions:
- Find a more suitable place for the first paragraph under the "Multi-Omics Integration Network Analysis" section in Materials & Methods. It reads more like an "Introduction" or "Results" paragraph.
- Table 1: In the column of "CO-MED", the Age at depressive onset starts from 0. Is that true? I don't know a newborn or an infant can be diagnosed with MDD.
- Table 1: In the column of "Adult onset (>=18 years of age)", the Age at depressive onset starts from 4. By definition, I thought this number should start from 18 something?
Reviewer 2 Report
The authors hypothesized that a multi-omics approach would derive distinct molecular signatures of early- versus adult-onset MDD. Therefore, the objectives of this study were therefore to 1) identify risk and protective single nucleotide variants (SNVs) associated with age at onset of MDD using a GWAS approach, and 2) integrate genomic findings with plasma metabolites to identify multi-omics differences between individuals with early- (<age18) versus adult-onset MDD.
These multi-omics networks enabled biological characterization of MDD by age at onset and provide a basis for future functional experiments aiming to investigate the mechanisms underlying the development of MDD across the lifespan. Derived early- and adult-onset MDD biosignatures show distinct associations between variants in/near INTU, FAT1, CNTN6, and TM9SF2 with plasma metabolites (phosphatidylcholines, carnitines, biogenic amines, and amino acids) for continued future investigation.
The study is well done. I found very extremely interesting the finding on spermidine. A recent study also found that spermidine could be anti-depressive drug (see, doi: 10.1007/s11357-020-00173-5)
Author Response
We thank the reviewer for their thoughtful critiques and insightful comments! Thank you for bringing the recent spermidine finding to our attention – we now include a reference to your suggested paper (Filfan et al., 2020) and additional work (Fiori et al., 2008) discussing the antidepressant roles of spermidine in the discussion section, as copied below (in bold) :
“The multi-omics networks also reveal additional associations for future mechanistic studies. This includes the associations of variation near CNTN6 with plasma carnitines, which are specific to adult-onset MDD. Such an investigation will add to the current characterizations of subgroups of MDD by acylcarnitine metabolomic profiles46. This also includes the associations of variants in FAT1 with spermidine, given that spermidine has recently been proposed as an antidepressant drug96,97.”
Reviewer 3 Report
Dear authors, it was a great pleasure for me to read your manuscript. The results are clearly presented and the conclusions are justified.
I have no concerns about the text and additional materials.
Overall this is an interesting paper that deserves to be published in the International Journal of Personalized Medicine.
Author Response
We thank the reviewer for their comments and your careful read of our manuscript. We are excited with the prospect of publishing in the international Journal of Personalized Medicine.
In summary, we appreciate the thoughtful comments from the reviewers and editor, and we have addressed each to the best our ability. We are pleased to submit a revised manuscript for your consideration and are thrilled with the prospect of publishing this work in the Journal of Personalized Medicine.